# Public Spaces, Equality, Diversity and Inclusion: Connecting Disabled Entrepreneurs to Urban Spaces

Helen Lawton Smith

Department of Management Birkbeck, University of London, London WC1E 7HX, UK; h.lawton-smith@bbk.ac.uk

**Abstract:** In the UK and in many other countries, the lack of support for disabled entrepreneurs is an economic, cultural, and societal issue. This is because while disabled entrepreneurs belong to and contribute to public spaces, there are often barriers to their full engagement in the local economy. Where interaction is well established, such entrepreneurs add to the cultural richness of places, to personal and societal well-being and economically by wealth creation. The goal of the study is to identify what can be done to overcome the marginalisation of disabled entrepreneurs, which leads to increased local equality of opportunity, thereby adding to the diversity of local economies and, thus, to a more inclusive society. However, as the evidence from this study of the geography of specialised networks which support disabled entrepreneurs in the UK shows, the entrepreneurial capacity of public spaces (inclusive entrepreneurial ecosystems) for disabled entrepreneurs is better in some places and entirely absent in others. It is this local dimension that has been missing in other studies of disabled entrepreneurs. By focusing on the formal networks that have been established to support disabled entrepreneurs rather than the entrepreneurs themselves, the particular knowledge gap that this paper addresses is the importance of the networks in making those connections and bringing about systemic change in urban spaces. They do this in three ways. They provide access to resources that disabled entrepreneurs need to start and grow a business; in turn, they need to engage with other local public and private sector organisations in order to sustain their own activities, and by their role as advocates on behalf of their members through their leadership, they increase the visibility of disabled entrepreneurs within urban spaces. The contribution to academic literature is to explore the interconnection between the agency of particular organisations to improve inclusive entrepreneurial ecosystems and overcome embedded exclusion within urban spaces. Examples from the UK and from the USA provide empirical insights into what can be done.

**Keywords:** equality; diversity and inclusion; disabled entrepreneurs; urban spaces; social spaces; public spaces; inclusive entrepreneurial ecosystems; advocacy

## 1. Introduction

How urban spaces affect the ability of organisations which support disabled entrepreneurs, or Entrepreneurs With Disabilities (EWD), to function and be connected to public urban spaces is a question with a number of dimensions. It raises economic issues because of the consequences of the undervalued and undersupported capacity for innovation and entrepreneurship by EWD. It is also a social issue because of sometimes adverse societal perceptions of the capacity of minorities to be fully entrepreneurial and achieve their personal and professional goals, thus missing out on societal beneficial cultural changes. The link to public spaces and equality, diversity and inclusion, therefore, relates to overcoming the marginalisation of disabled entrepreneurs. Where this is addressed by different actors within public spaces, it leads to increased local equality of opportunity, thereby adding to the diversity of local economies and a more inclusive society.

The goal of the study is to identify what can be done to overcome the marginalisation of disabled entrepreneurs, which leads to increased local equality of opportunity, thereby adding to the diversity of local economies and, thus, to a more inclusive society. The context

is the challenges faced by entrepreneurs with disabilities: funding and investment, lack of support system, skills, social prejudices, network and connection, health and physical access [1]). This, in turn, impacts individuals' well-being and, thence, their happiness [2].

These issues are important because public spaces have been described as places where "human exchanges and relationships, the diversity of use and the vocation of each place and the conflicts and contradictions of society are manifested" [1]. Therefore, while public spaces are also physical spaces, the emphasis in this paper is on the social public space. The support networks that are discussed work best for EWD because they are social and involve social interaction, social mixing and inclusion.

However, there is also a relevant physical dimension in urban centres. For example, disabled people may need local physical support, for example, in access to online websites or face-to-face interaction (see SAMEE case study below). They may also benefit from role models, and it can be easier to identify with local role models. Both kinds of interventions on behalf of disabled entrepreneurs may change the nature of the relationships and discourse [3] within public spaces by building social inclusion where hitherto there has been exclusion.

This paper provides evidence on how urban places affect interaction through the actions of specialised networks within Inclusive Entrepreneurial Ecosystems (IEE) [4], which are the source of and conduits for entrepreneurial support. It answers the question, "How are disabled entrepreneurs embedded in public spaces?" The focus of this study is on how formal, specialised networks' ability to function is affected by opportunities, or their lack, within urban spaces. Evidence is drawn from a UK study of geographies of support for disabled and ethnically diverse entrepreneurs (2020–2021). It is shown that in some places in the UK, there is close interaction between networks and other local organisations, but more often, there is a picture of activity existing in isolation from other actors in urban spaces.

A point of reference is a research report [5], which analysed a cross-city learning collaborative designed to foster inclusive innovation and entrepreneurial development in four US cities—Cleveland, Detroit, Durham and New Orleans Forward Cities, which began in 2014. This is an example of a coordinated approach to building local urban inclusive entrepreneurial ecosystems designed to develop connections between urban spaces changing perceptions, dialogues, interconnections and cultures. These are processes rarely discussed in the literature [6]. The initiative was designed to strengthen local Entrepreneurial Ecosystems (EE) by bringing together a broader group than just those players usually involved in "inclusive innovation" discussions.

The advantage of analysing using the place-based Entrepreneurial Ecosystems (EE) concept [7,8] such as in both Forward Cities and in the research reported here, is that it can be used to assess the current opportunities and challenges in particular public urban places. However, as the ecosystem idea originally focused on high growth, it has to be adapted as it tends to obscure the diversity of entrepreneurship that exists within entrepreneurial ecosystems [9,10]). Therefore, a better understanding of how disabled entrepreneurs access resources and support within EEs [11]) is needed. The assumption that all entrepreneurs have equal access to resources, support and success outcomes within an ecosystem per se and, in particular, public spaces rarely holds in practice [12,13]. Like other entrepreneurs, disabled entrepreneurs range in activity from self-employment in a craft-based sector to advanced engineering. These points are particularly relevant to urban planners who need to take into account the specificities of the physical and social needs of disabled entrepreneurs.

To overcome this conceptual limitation, a development, that of IEE, reflects the need for a more sensitive approach when addressing minority entrepreneurship. The IEE concept is used to interpret both the uneven embeddedness of disabled entrepreneurs in urban spaces and the possibilities of improvement through more inclusive interaction involving the sharing of experiences and knowledge. This study's contribution is conceptual and empirical by exploring what makes for local differences and how these are explained

through the intersection of urban spaces as social spaces within an IEE framework. In other words, the focus is on how change happens or is constrained within urban spaces.

The next section reviews how disabled entrepreneurs' embeddedness in urban spaces can be interpreted. The section following reviews the methodology of how evidence was gathered on this phenomenon. The data are then analysed together with a case study of a network which supports disabled entrepreneurs. Finally, some conclusions are drawn.

## 2. Materials and Methods: Disability, Place, Public Spaces, Urban Centres and Inclusion

### 2.1. Introduction

In order to contextualise how disabled entrepreneurs' relationships are mediated by the formal networks to which they belong within particular urban spaces, it is necessary to focus on what being a disabled entrepreneur means in practice.

Defining what is meant by disability helps position how and why social inclusion within entrepreneurship is an economic activity within urban spaces. Disabilities are extremely diverse [14] and are not a fixed characteristic of individuals; that is, disability is often a temporary condition rather than a permanent status. Popular stereotypes of disabled people as being permanent wheelchair users or as blind from birth persist, shaping public perceptions and informing policy approaches [15].

Definitions of disability, and the policy approaches that they inform, are shaped by two contrasting conceptions: the medical model of disability and the social model. The medical model treats disability as a characteristic of the person. In contrast, the social model of disability, pioneered by Oliver 1990 [16], assumes that people are disabled by societal attitudes, institutions and environmental barriers. The social model distinguishes "impairment"—limitation of the mind and body—from "disability"—social exclusion [17]. These distinctions are important because they are likely to influence who is defined as disabled in particular places, with implications for eligibility for support in the publicly-funded active labour market and entrepreneurship programmes [15].

The link to the social model of disability and entrepreneurship lies in three particular characteristics of public spaces [18] in the relationship between urban structures and the culture of usage of space. The first is physical features, including the shapes and forms defining the space. The second is the distribution and behaviour of users, which reflect social order. The third is the flow of human movement which relates to the first two.

Most relevant here are the first and second, which relate to the behaviour of users, interpersonal communication and social activities [19], as well as physical spaces. This is because disabled people may have non-standard physical and social needs, which affect how and the extent to which they are included in or omitted from networks of economic activity in urban places. These can be physical spaces or social spaces. The first includes issues of a lack of awareness and understanding of accessibility, both physically and virtually, which affects disabled entrepreneurs' experiences of accessing urban spaces.

The second is that, as with open spaces, social spaces are sites of social interaction, social mixing and social inclusion. A public space provides an arena for the exchange of ideas, friendships, goods and skills [2]. This is also relevant to another dimension—that of the cultural use of space. As the paper shows, culture, in this case, means where social inclusion is more advanced in some places than others. Of relevance here is a study by Agboola et al. (2017) [2], who argue that residents' well-being reflects the experience within the interplay of individuals' and groups' social interactions.

This idea is captured in this quote, "the more diverse and lively urban spaces are, the more equal, prosperous and democratic society becomes". This assertion is based on the very definition of public space as "an open, freely accessible and democratic environment" [3] (Rogers cited). Public spaces, however, can be shaped by struggles between different ideologies, discourses, political decisions and daily activities. These can take place at personal, interpersonal, local, national, supernational and global scales [3]. Of particular relevance here are cultural, that public spaces allow for or inhibit new perspectives on the

diversity of economic activity and how national policy influences potential resources that networks can help their members access in local public spaces.

The concept of IEE is a way to link the physical and social dimensions of urban spaces by focusing on systemic elements of entrepreneurial interaction, which highlight features of inclusion and exclusion.

### 2.2. Inclusive Entrepreneurial Ecosystems

The context to place-based analysis of inclusion or exclusion in interactions is that urban places differ in their mix of populations by age, ethnic minorities and disabled people. Institutionally, places also differ in the number and type of organisations that are actually or potentially available to provide support roles and which also differ in the outcomes that they seek. Intrinsic to entrepreneurial ecosystems are networks established among the diverse stakeholders' impact on the configuration, evolution and outcomes of entrepreneurial ecosystems [20]. This includes possibilities of public policy intervention. For example, in the UK, municipalities have autonomy in regulating space and the urban environment and hence can make choices about whether or not to focus on more inclusivity. In the Forward Cities programme in the USA, the decision was made to change discourses and actions to overcome the marginalisation of ethnic minority entrepreneurs.

In other words, places vary in the types of organisations that are or might be agents for change. Intrinsically, it is an issue of governance factors that underpins how quality interactions that compose an entrepreneurial ecosystem develop and change over time [21]. At the heart of the concept is the interconnectedness and social nature of EEs [9,10,12] that are manifested in urban spaces.

An interdependence between urban context and the potential for economic development has been observed for disabled people [22]. This comes from specific economic, social and cultural framework EE conditions [7] (see Figure 1). This includes whether particular locations are network-rich or network-poor [23]. It is also a cultural issue since culture (positive or negative) has an influence on entrepreneurship and on support for entrepreneurship. This influence may persist in the long term [24].

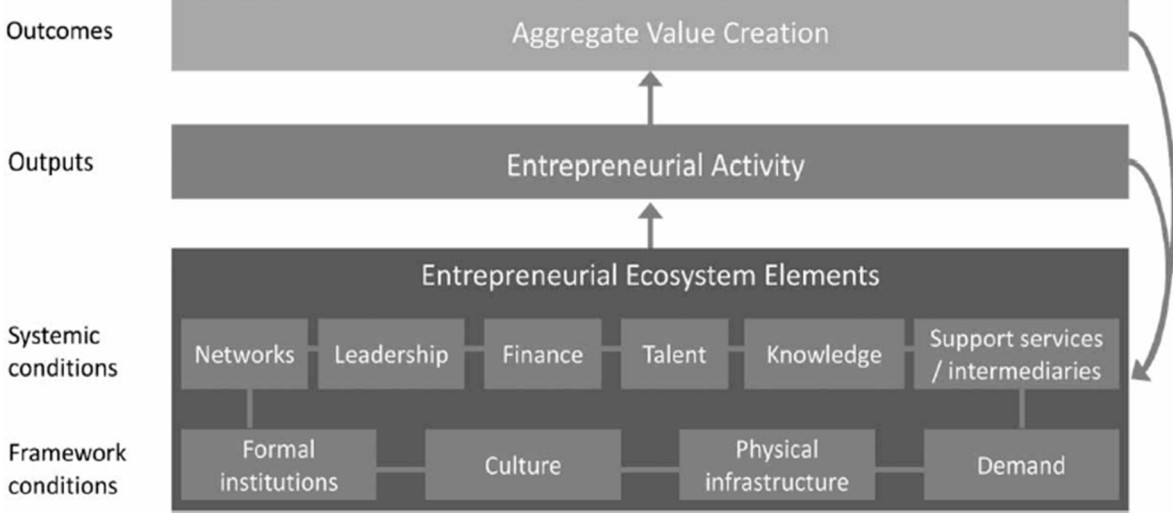

**Figure 1.** Entrepreneurial Ecosystem model. Source: ([7], p. 1765).

Under certain conditions, networks based on the geographical dispersion of communities enable higher levels of business competitiveness by facilitating access to resources and markets by minority [25]. Chowdhury and Savasthasy (2022) [26] developed a theory of marginalised stakeholder-centric entrepreneurship in order to analyse why and when this is problematic. Their focus is on inclusiveness, explaining how existing firms can utilise marginalised stakeholders, for example, disabled entrepreneurs. This can be by customer

firms employing a variety of ideas, resources and interactions with marginalised stakeholders. Firms can then filter, internalise and finally realise important elements that improve a variety of related socioeconomic, ethical, racial, contextual, political and identity issues. This idea can be utilised to highlight how an IEE could develop by addressing marginality issues, for example, for disabled entrepreneurs by introducing a different perspective on the agency for more inclusivity as an output of entrepreneurial activity.

The EE builds on traditions in studies of urban spaces such as industrial districts, clusters and learning regions. These are all versions of urban spaces. Entrepreneurs are central actors within the ecosystem [27]. In the EE approach, entrepreneurial actors not only play a key role within the system, but they also have a role in creating and sustaining the system over time [7]. However, when marginal populations which are characterised by a lack of network centrality [28] are examined, there is a shift in the primacy of entrepreneurial agency towards other frameworks and system conditions of EEs.

This is the conceptual advance of the IEE approach [4]. It contributes to broadening the conventional entrepreneurship paradigm by placing a particular set of entrepreneurial actions and actors at the heart of the concept. The IEE system also includes actors not found in standard entrepreneurial ecosystems but relevant to the embeddedness of disabled entrepreneurs in urban spaces. These include specialist networks, volunteers, charities and not-for-profits whose remit is to support minority entrepreneurship. An IEE approach specifies the need for greater recognition of the role that sociocultural norms, formal institutions and hierarchical structures play in determining minority entrepreneurs' access to resources located in urban spaces and, thence, more extensively.

In the inclusive EE approach, four of Stam's systemic conditions are prioritised to help understand how interactions in urban spaces happen through networking (one of the six conditions which is a given in this context). These are leadership, finance, knowledge and intermediaries (in this case, specialised support networks). Finance, the issue of sustainability, is often a pressing concern for specialised support networks. It is intermediaries who potentially provide leverage between all of those elements. These are not considered to be formal organisations and are part of the framework conditions. In addition, not covered by the Stam framework is the role of advocacy in driving forward the inclusivity agenda and local and national policy frameworks.

While leadership can be important for identifying and coordinating responses to weak ecosystem links [29], advocacy overlaps with the concept of leadership. In this analysis, we highlight how specialised formal networks, professional support organisations and purely web-based initiatives have an important advocacy role in highlighting the economic and societal value of their constituencies as well as the challenges that they face. Advocacy means that messages are publicly articulated, and then leadership requires that change or at least action ensues. Advocacy often has an urban space connection as it is an articulation of sets of physical and social conditions that present barriers to economic and social inclusion. It is designed to influence public opinion as well as political action at local and national levels (see Sadri, 2017 [3] on this point).

To illustrate how change can be designed within urban spaces through interactions designed to broaden inclusivity within an IEE framework, leadership was provided by Friends of New Orleans. Each city had an Innovation Council, each made up of some 35 volunteer entrepreneurs, funders, corporate leaders and representatives from economic development and business support organisations. In the first stage of the initiative, Innovation Council members identified neighbourhoods or corridors where they would identify new ways to direct and coordinate resources in order to increase entrepreneurial and small business activity and connectivity. Key coordinating activities were "Convenings" consisting of both physical and virtual events such as topical panels, tours and formal networking.

The initiative recognised the need to address inequalities in opportunities for minority entrepreneurs, who have been described as "marginalised stakeholders" by Chowdhury and Sarasvathy (2022) [26]. Minority entrepreneurs face additional barriers to overcoming

disadvantages of ethnicity or disability in accessing entrepreneurial resources (finance, mentoring, premises, employees). The selected areas were all located in underserved areas with the potential for entrepreneurial growth and small business development. It was intended "to accelerate, connect, and concentrate local work toward more inclusive innovation and entrepreneurial development" (page 7) in "communities of color".

To summarise, the interpretation of urban spaces as physical and social spaces which are inclusive of disabled entrepreneurs is through an IEE lens. The focus is on the agency of a particular kind of organisation, networks which support disabled entrepreneurs, to bring about more inclusivity in the culture of the use of space [18].

*2.3. The Study*

The data analysed in this paper are based on a research project (2020–2021) funded by the Regional Studies Association. The purpose was to identify geographies of formal support in the form of specialised networks for disabled and ethnically diverse entrepreneurs in the UK. The objectives were to map the networks with a view to identifying regional differences in their availability and then to find out what difference location made to how the networks functioned as intermediaries. This project built on evidence from a previous Innovate UK (the UK government's innovation agency) project (2019), which identified the barriers, challenges, opportunities and support needed for ethnic minority and disabled entrepreneurs (non-gender specific) participating in business innovation [30]. The gap in this study was that the geography of support for those groups was largely absent.

The new research was carried out in two main stages. The first involved a mapping exercise designed to identify by UK region (including the devolved administrations: Wales, Northern Ireland (NI) and Scotland) all dedicated networks which specifically support ethnic minorities, disabled entrepreneurs or both groups.

The exercise also included identifying all national networks and organisations, as well as government-based initiatives designed to support entrepreneurship and innovation among the two groups.

Various sources and search processes were used to identify networks. These included using a snowball technique of recommendations from participants in the previous study, including the Innovate UK project Advisory Board and government agencies. In addition, web searches were conducted to identify networks and data on universities supporting the two target groups was collected. In total, 77 organisations were identified, 64 of which responded to requests for information and/or interviews.

The second stage involved conducting interviews with networks, individual disabled and ethnic minority entrepreneurs, universities and policy bodies. The sample relating to disability entrepreneurship included 10 networks, 9 policy and/or parliamentary bodies, a national disability organisation, 2 business organisations/trade bodies (e.g., Federation of Small Businesses), 5 academics and 4 other regional bodies (31 in total). Of the ones which did not respond, two networks specialising in disabled entrepreneurship, two devolved region organisations, four policy-making/advisory bodies and two trade bodies.

The interviews gathered information on the organisation's activities with urban spaces and wider geographical engagement, for example, with public policy bodies, evaluation of performance and sustainability. Interviewees were also asked for policy recommendations. While interviews were conducted in the majority of UK regions, it did not prove possible to conduct interviews with networks in Scotland or in the South East. More networks came to light during the later stages of the project. They were not contacted for an interview.

The interviews focused on understanding the extent to which urban location is a factor in what they are able to achieve and the challenges they face in providing support. Other interviews were also conducted with some of the national policy-making bodies, such as Innovate UK (which is the national government's innovation agency) and the secretariats of the All Party Parliamentary Group for Inclusive Entrepreneurship (APPGIE). To triangulate the interviews' analysis, secondary data such as archival data, news and public reports were gathered and analysed in each individual case.

In order to facilitate the data analysis, interviews were recorded and transcribed. The transcripts were then thematically analysed [31]. Computer-based analysis for this research was used using NVivo 12 Plus. The researchers first identified a group of principal categories of data related to the lines of inquiry of the study. This is in line with Strauss and Corbin's (1994) [32] argument that a priori definition of certain baseline concepts in a study supports a grounded theory approach to data gathering and analysis, which is applicable for an interpretive portrayal of the world [33].

The a priori codes are codes that are developed before examining the data (interview transcripts). Some of the identified open (initial) codes included:

-   Organisation's activities and excellence
-   Lack of public policy initiatives
-   Lack of BAME or disabled support
-   Stakeholder and beneficiary features
-   Effect of location and London centrism

The networks identified in the study are shown in Table 1.

**Table 1.** Geographical distribution of networks and organisations that support disabled entrepreneurs.

| * Unclear If It Still Exists | Disabled Entrepreneur Support | Summary |
|---|---|---|
| Scotland | - Work4Me | |
| Northern Ireland | - Disability Action<br>- UnLtd | - Regional effect: Northern Ireland is one of the poorest regions within the United Kingdom. Only disability support organisations were found.<br>- British Business Bank is facilitating the setting up of a Northern Ireland Equity Taskforce.<br>- Disability Action Exploring Enterprise Programme in the North Down area. Provides support for disabled people who need some support in gaining employment or becoming self-employed.<br><br>UnLTd provides funding to support social entrepreneurs and people with innovative solutions with the potential to change society for the better. UnLtd particularly encourages applications from people with disabilities. |
| Northeast England | | |
| Northwest England | - Association for Disabled Professionals * [Bolton]<br>- Universal Inclusion [Lancashire] | ADP members consist mainly of disabled people living in the UK (all of Executive Committee are disabled), Universal Inclusion and the Inclusive Entrepreneur Network have created a holistic membership organisation both digital and real world comprising of business support, bespoke inclusive events and PR opportunities, peer support, advocacy and mentoring for people who have protected characteristics, particularly disabled people. Its successful inclusive entrepreneur programme combines traditional start-up and scale-up elements alongside an essential health and well-being strand. Universal Inclusion provides Secretariat to the All Party Parliamentary Group for Inclusive Entrepreneurship, and members of the Network feed directly into the APPG. Universal Inclusion and the Network collaborate with cross sector organisations including academia to create an international inclusive entrepreneurial ecosystem. These organisations are based in Edgworth, Bolton but operate throughout the UK and internationally. |

Table 1. *Cont.*

| * Unclear If It Still Exists | | Disabled Entrepreneur Support | Summary |
|---|---|---|---|
| Yorkshire and Humberside | - | Bradnet * | Bradnet (now part of Inspired Neighbourhoods) had an initiative to support disabled entrepreneurs but now provides care and support. This means that dedicated support organisations for disabled entrepreneurs are *missing* in Yorkshire. |
| East Midlands | - <br> - | Disability Direct * [Derbyshire] <br> Blind Business Association Charitable Trust (BRACT) [Northamptonshire] | The Enterprise Centre was part of Disability Direct but now have no contact with disabled innovators. Had 3 years European Social Funding some 10 years ago. <br> BRACT aims to maximise the potential for long-term business success and offers business advice, mentoring, seminars, conferences, need-orientated projects and a small grants fund. |
| West Midlands | | | |
| Wales | - | Disability Wales: Endeavour Crowdfund | Business Wales offers an accelerator programme designed to champion diversity in entrepreneurship in Wales and aims to develop participants' core business skills and a "success mind-set" through webinars and masterclasses, one-to-one expert mentoring and coaching, from inspirational speakers, role models and business growth experts. <br> Disability Wales' Endeavour project seeks to inspire and support disabled entrepreneurs to establish their own businesses. |
| East England | - | Accessful Foundation (online only) [Essex] | MENTA offers affordable training, advice and key services to aspiring and established business owners in Norfolk, and Suffolk supports both ethnic minority and disabled entrepreneurs as part of their broad portfolio. Not-for-profit. <br> Accessful Foundation aims to make entrepreneurship accessible for everyone by facilitating networking and mentoring, by giving grants, by creating and promoting representation and by being a transparent, diverse and innovative charity that campaigns for positive change. |
| Southeast England | - <br> - | Disabled Entrepreneurs Network (DEN) <br> Disability Dynamics [Hampshire] | Disabled Entrepreneurs Network (DEN) is operated by the Association of Disabled Professionals (ADP). DEN provides networking opportunities and information services for self-employed disabled people and those setting up and running their own small businesses throughout England, with regional contacts in each region. <br> Disability Dynamics is registered in Hampshire and was established in 2008 to support disabled entrepreneurs. |
| Southwest England | - <br> - | SAMEE [Bournemouth] <br> Mutually Inclusive [Bristol] | - SAMEE supports self-employment and provides enterprise support for disabled entrepreneurs, from pre-start-up to existing businesses through business advisors, signposting to other organisations and careers advice. <br> - Mutually Inclusive offers mentoring, advocacy, enterprise support and support for people assisting disabled persons, especially people with learning difficulties dealing with Access to Work. |
| London-based and National Organisations | | | |
| Consulting/Mentorship/ Capacity Building /Training/private enterprises | | | |

**Table 1.** *Cont.*

| * Unclear If It Still Exists | Disabled Entrepreneur Support | | Summary |
|---|---|---|---|
| Networks/Networking/ Alliances | - | Learning Disabilities | Learning Disabilities In Business programme explored the route to self-employment and small business ownership for those people with learning disabilities interested in an alternative path to work. |
| Acceleration/Incubation/ Co-working Space | - | Global Disability Innovation Hub | Global Disability Innovation Hub is a research and practice centre driving disability innovation for a fairer world. |
| Charity/Grants/Awards (non-profit) | - | Leonard Cheshire Stelios Awards for Disabled Entrepreneurs | Leonard Cheshire Stelios Awards offers annual financial prizes for disabled entrepreneurs. |
| Funding/VC (commercial) | - <br> - | Kaleidoscope Investments <br> Disabled Entrepreneurs | Kaleidoscope Investments provides Business Investment supports to disabled entrepreneurs to develop businesses. Disabled Entrepreneurs is the charity arm of Kaleidoscope Investments and offers business support including one-to-one or collective business mentoring sessions. |

**\*** This organisation no longer appears to exist.

## 3. Results

In order to place the data collection elements in the study into a geographical context, evidence shows that the highest percentage of self-employed people with disabilities in the UK are in some of the more urban places. These are the South East (20%) followed by Greater London (13%). However, South West England, which is largely rural, has a 12% share. The areas with the lowest are Northern Ireland (2%), North East England (4%) (which is very urban), East of England (4%) and Wales and Scotland, both 6% (IPSE, 2019). However, according to Disability Action, 1 in 5 people in Northern Ireland have a disability.

More than seven million people aged 16–64 of working age are classified as disabled under the UK Equality Act 2010. Over a seventh (16%) are self-employed as their main job (14% sole traders). Of the self-employed, three-quarters (78%) describe working for themselves, and 1 in 5 (19%) identify themselves as running a business or professional practice. The ratio of 3:2 males to females is consistent with UK figures. However, the proportion of self-employed women is increasing. These trends are more likely to be caused by greater "necessity entrepreneurship" or redundancy for men than for women (IPSE, 2019). IPSE report that older people are more likely to be disabled. Those in the age group 50–69 account for 28% and those 60+ for 26%, meaning that those aged 50 and over account for over half of disabled people. Another quarter is between 40 and 49.

The geographical context to the challenges faced by place-based inclusive entrepreneurial ecosystems, therefore, includes an uneven distribution of disabled entrepreneurs. This context has implications for demand for support, particularly for a key EE system element, finance. A point to note is that location, hence in urban spaces, is an important factor in success, irrespective of ethnicity, disability, age and gender. For example, Greater London is the toughest place in the UK to be an entrepreneur, with just 71% of business owners in London reporting a profit in 2019. Meanwhile, entrepreneurs in the South East and North East see the most success. Differences between London and other areas of the UK are linked to a higher density of start-ups and tougher market competition. Higher costs of living and operating, and greater disparity between poorer and wealthier neighbourhoods, also help explain these findings [34].

### 3.1. The Mapping Exercise

The data showed considerable regional, hence local variations, in public spaces in the presence of networks established to support disabled entrepreneurs. Several regions, therefore, have no IEE elements in the form of intermediaries which support disabled entrepreneurs. In some regions, for example, in NI and Scotland, there was limited activity.

In the North East, disabled entrepreneurs appear to not have local access to specialised support. The lack of networks may be a reflection of the fact that only 4% of the region's self-employed have a disability. The only network that supported disabled entrepreneurs has ceased that activity following a withdrawal of funding.

In Wales, a government agency, Business Wales, leads on activity including funding for minority entrepreneurs, and the Welsh national organisation, Disability Wales, which has some 550 members, provides a key coordinating role in supporting disabled entrepreneurs.

Some networks offer national coverage. For disabled entrepreneurs, examples include the Inclusive Entrepreneur Network, the Association of Disabled Professionals (ADP) and the Kaleidoscope Investments (Disability) Awards. Some networks are online only, for example, Accessful Foundation. However, being online does not necessarily mean that there is not a local focus; hence social spaces of interaction can be both local and non-local.

There are also a dozen or so national business-based initiatives, such as for disabled entrepreneurs. Examples are the Royal National Institute for the Blind, working in partnership with the British Business Bank [4], the Disabled Entrepreneur–Disability UK network [5] and the Leonard Cheshire Stelios awards.

With reference to IEE systemic conditions, three main categories of intermediary organisations were identified. These are membership organisations, charities and commercial organisations. In all cases, the primary activities are networking, mentoring, advocacy and influencing policy. A consequence of this relative absence is that those networks which support disabled entrepreneurs take on more geographically extensive activity building on experience in their core urban space. This will be illustrated by the case study of the SAMEE project.

### 3.2. The Interviews

Of relevance to this analysis, interviews focused on the relevance of place in how specialised networks were able to function in providing services to their members by engaging with local potential EE actors and stakeholders, the impact of place on how individual entrepreneurs were able to access support and the policy context. Each network was asked to identify actual and potential local partners which could be approached to provide extra resources. National policy-makers were asked about their agenda and programmes in providing access to support for the networks and their views on a place in policy-making.

The six prioritised themes in this analysis relate to the extent to which urban spaces affect interaction through specific forms of networking, in other words, in contributing to how public social spaces evolve. Four are those highlighted above from the Stam (2015) [7] framework: leadership, finance, knowledge and intermediaries (in this case, specialised support networks that potentially provide leverage between all of these elements). In addition, are two more from the IEE approach [4]. These are the role of advocacy in driving forward the inclusivity agenda and local and national policy frameworks.

Next, some general trends are given with respect to the six aspects of the IEE approach leadership, finance, knowledge and intermediaries, advocacy and local and national policy frameworks. This is followed by the case study of the SAMEE project, which illustrates all of these themes in relation to urban public social spaces.

In the absence of formal leadership, which was present in *Forward Cities*, specialised minority entrepreneur support networks provide local urban leadership by identifying and providing appropriate business support for their members. Services include mentoring, networking and helping to access government funding. Many networks, for example, those for disabled entrepreneurs, offer health and well-being support which enables en-

trepreneurs to take on and manage entrepreneurial activities. Networks provide both the mechanisms of support and the basis for the representation of minority interests at the local level. They also link localities to national and sometimes international agencies. Examples are included in the case studies below.

In the *Forward Cities* programme, an Innovation Council brought together local organisations that already had or were interested in supporting minority entrepreneurship in particular urban spaces. In the UK, it was the specialised networks that fulfilled that coordination function linking the entrepreneurs to urban spaces by increasing their visibility and legitimacy as entrepreneurial actors. Their Boards of Directors, which include entrepreneurs, big businesses, city councils and universities, in many cases provide the equivalent of innovation councils.

The mix of the kinds of organisations which were engaged in providing support through the specialised networks varied. For example, support from local authorities in some places was strong and in other places absent. The commitment of such organisations as the Chambers of Commerce and Federation of Small Businesses (FSB) to supporting disabled or ethnically diverse entrepreneurs is also context-dependent. Three examples are given of the ways and extent to which Networks act as intermediaries connecting entrepreneurs to urban places and increasing the well-being of disabled entrepreneurs.

### 3.3. Funding and Sustainability

Sustainability is an ongoing key concern for the majority of networks. While many organisations have been founded recently, a number of networks have ceased their support for minority entrepreneurs due to a lack of funding.

An example of a network which has not survived is one from Yorkshire and Humberside. In the early 2000s, the network that supported disabled entrepreneurs was part of the Local Economic Growth Initiative (LEGI) in Yorkshire. It helped people become more entrepreneurial. A consortium of stakeholders was established, including the local Chamber of Commerce, the City Council and other independent sector organisations. The LEGI programme lasted a couple of years, but the entrepreneurship arm lasted 9 years.

While this was an example of collaboration between local stakeholders around a common goal of supporting disabled entrepreneurs, it did not survive, leaving a policy gap. The gap was identified as the need for Northern authorities to raise "the business can do"—a purposeful agenda. The city of Bradford was said to be poor at proposing agile initiatives, compared to its neighbour Leeds which was more go-ahead. There was very little new money, and local authorities needed to embrace an enabler role in helping micro-businesses through a range of support, including infrastructure support, advice, short-term catalyst support, bringing skills onto company boards and providing help with applying for business support. A significant barrier was the lack of new thinking by organisations such as the Chambers of Commerce and local authorities. Success stories of disabled entrepreneurs would highlight those initiatives which are disability-led entrepreneurial organisations.

### 3.4. Access to Knowledge

Rather different in IEE compared to the basic EE model is that, in the diversity field, universities are increasingly becoming local actors in inclusive entrepreneurial ecosystems. In some cases, they give active local support to minority entrepreneurs as well as giving credibility to specialised networks and entrepreneurs in both groups by association with their brand and their research. This is also reciprocal as universities benefit from association with practitioner best practices and from the potential impact of their research. While the former effect tends to be local, the latter is often national and international.

An example is in the North East, Northumbria University's Business Clinic. It provides free consultancy advice to SMEs, multi-nationals and not-for-profit organisations who are looking to grow by taking their business in a new direction, explore new challenges or require fresh eyes to help them succeed. The service is provided by final-year undergraduate and postgraduate business students with support and guidance from teams of experts at

Newcastle Business School. It also works in urban spaces working with charities supporting entrepreneurs and others with disabilities. Latest projects include encouraging/supporting autistic entrepreneurs to establish and run their own businesses, identification of barriers and recommendations to overcome discrimination in the recruitment of individuals with autism and a feasibility study into appropriate real-world employment opportunities for young people with learning disabilities.

The clinic is well-connected locally and nationally to organisations that provide support, having well-established engagement with local organisations, including the North East Local Enterprise Partnership; the North East England Chamber of Commerce; UnLtd, North East Office (Bradford); Santander UK Business Banking Office, North East Office; NatWest Bank, a commercial bank and the British Business Bank, a government-owned business development bank dedicated to making finance markets work better for smaller businesses.

### 3.5. Advocacy

Table 1 gives many examples of advocacy activity by specialised networks. They include Scotland's Radiant and Brighter Futures Women's Leadership and Enterprise programme, Universal Inclusion and the Inclusive Entrepreneur Network in the North West. The interpretation here is that social spaces change as dialogues begin to make an impact on cultures of inclusion and exclusion.

### 3.6. Policy Agenda and Public Spaces

A very clear pattern that emerges is that policy, business organisations and academic interest in disabled entrepreneurs locally and nationally are much more recent and are at different stages in development than that for other minority entrepreneurs.

In some parts of the UK, those with subnational tiers of government, action is being taken, for example, in Northern Ireland and Wales. Enterprise Northern Ireland (ENI), the enterprise agency, has a commitment to EDI and to inclusivity in urban spaces: "Enterprise Northern Ireland's primary aim is to grow the economy and enrich local communities through development of enterprise and entrepreneurship. All activity is importantly underpinned by a commitment to inclusivity as future growth of our society will succeed only if the rich diversity of our people and their entrepreneurial spirit is allowed to flourish". It has developed an ENI Equity Taskforce, which will develop a set of recommendations for action.

Where Wales differs from Scotland is in the active stakeholder involvement of national organisations, including Business Wales (Welsh government) and the charity Disability Wales, which is the national association of disabled peoples' organisations in Wales. In 2020, Disability Wales launched a programme, the Disability Wales Endeavour fund for disabled entrepreneurs (crowdfunding, training and coaching). The project seeks to inspire and support disabled entrepreneurs to establish their own businesses.

Over the life of the project (2020–2021), IEE ecosystem changes have included the development of a more holistic and inclusive national-level policy approach to minority entrepreneurship. Innovate UK, the Department of Business, Economics and Industrial Strategy (BEIS), UK Research and Innovation (UKRI) and the Cabinet Office Disability Unit have all made published commitments in policy and practice to EDI. For example, in January 2022, Innovate UK organised "Innovation without limits: increasing disabled innovators access to Innovate UK programmes and funding", to which a wide variety of stakeholders, including academics, were invited, including the lead author of this paper.

Part of the policy landscape in this context is the APPGIE. This is very active in identifying where policy needs to be improved better to support the interests of these groups of entrepreneurs. However, as of yet, the Disability Unit of the Central Government Cabinet Office is the only one to have adopted a regional approach to engaging with local groups: its regional stakeholder committee.

A recent trend is for policy-makers, practitioners and academics to join forces to investigate the challenges in developing and delivering effective policy that is sensitive to the needs of minority entrepreneurs. However, this is patchy.

Next, the example of the SAMEE project illustrates the theme of urban spaces as physical and social spaces affecting how disabled entrepreneurs are connected to urban spaces through the formal networks that have been established to support them.

The network is a charity based in the city of Bournemouth in South West England. It was founded in 2016 as a charitable organisation with a mission to alleviate poverty for disabled adults by narrowing the disability employment gap. It has taught self-employment skills to enable over 200 Dorset-based disabled adults to gain further independence by generating their own income. It does this by supporting disabled adults to explore non-traditional forms of employment that will fit around their health challenges and reduce the barriers they face to employment opportunities. The network contributes to entrepreneurs' personal happiness by building personal entrepreneurial identities and provides opportunities "to develop confidence and achieve goals and transform lives". It reinforces the local urban IEE through peer-to-peer mentoring, marketing, developing skills and viability of a business idea. Network graduates work as mentors, thus reinforcing the local IEE.

As an employment centre, it has created a "safe space" which has free internet and can offer help with job applications. It can help with skills, help with networking and even offer voluntary work. For those wishing to become self-employed, it will offer advice such as how to organise their time and book-keeping. Everything in this social space is disability-friendly, including braille, hearing loops and emotional support.

From its starting point, it has extended from the local to trans-regional, now having a geographical coverage of Dorset, Devon, Hampshire, Somerset, Wiltshire, Oxfordshire, Berkshire and the Thames Valley, with over two-thirds still based in Bournemouth in Dorset. It expanded because there were gaps in support in the other areas.

As an intermediary, it has an extensive network of local stakeholders, including local charities, for example, Dorset Mindfulness, organisations with specialisms in disability, and the Dorset Chamber of Commerce, which works alongside the Local Enterprise Partnership. The network has access to the local city council through the Chamber of Commerce and to the police and transport offices. However, it does not engage with the local FSB, which sees itself as a national organisation and does not have local knowledge or an interest in small start-ups. "Our guys aren't big enough for them", and "the fees too expensive". The network has some good contacts with Bournemouth University, which supports some of their events.

However, this network's sustained engagement in this urban space and as a social space is not guaranteed. Within its core urban space, organisations such as local authorities lack funding to support the charity. The network believes that if it were in London, it would get more funding. A London location would offer more prospects of raising finance. "if I could have an office in London I would be a happy girl". There is a tendency for the public perception of the South West to be that it is an affluent area. In part, this is because it is where many Londoners have their second homes. However, there is extensive deprivation in the area.

The need for greater financial support is illustrated:

*"And at the moment, the government nationally does not have any programme or idea of business support . . . And what we should be doing is having a three-year programme, which is non-government biased, which is then having its own local nuances, and where we need to drive certain effects into, into the country. And right now, there's a lot of people going to be looking at self-employment, how we do that within our locality areas, because Cornwall will be very different from London".*

This network brings visibility and support through local and national sponsorship to urban spaces embedding them in the local IEE. It also sees opportunities for advocacy and 'disturb status quo thinking' so as to bring about cultural as well as political change. The

CEO is on the regional stakeholder committee of the Cabinet Office Disability Unit. She is a leader and an advocate for providing local and national role models.

## 4. Discussion and Conclusions

This paper uses data and findings from a recently completed study of network organisations which support disabled entrepreneurs in the UK and from an initiative in the USA. The question raised is as follows: How are disabled entrepreneurs embedded in public spaces? At issue is how do urban spaces influence how networks connect disabled entrepreneurs to public urban spaces. This, in turn, relates to how public spaces can be better sites of equality, diversity and inclusion and diverse social spaces. The goal was to identify what can be done to overcome the marginalisation of disabled entrepreneurs, which leads to increased local equality of opportunity, thereby adding to the diversity of local economies and a more inclusive society.

The mapping exercise of the geography of support for disabled entrepreneurs in the UK showed considerable variations in the presence of local support in urban places. In some parts of the UK, only online support is available. This matters because disabled entrepreneurs may have non-standard physical and social needs that require specialised support.

Of note is evidence from the study of how the agency of networks can lead to overcoming the marginalisation of disabled entrepreneurs. What happens is that in some public places, entrepreneurial capacity is helped by improving inclusive entrepreneurial ecosystems through providing localised specialised support in conjunction with incorporating the interests of local (and national) political and commercial organisations (see, for example, the SAMEE project example). The network has created "a safe space" for people with disabilities looking to become self-employed.

In principle, the agency works on behalf of the marginalised stakeholders within public spaces, which leads to increased local equality of opportunity for particular groups of entrepreneurs. They thereby add to the diversity of local economies as well as improve the well-being of disabled entrepreneurs [2]. However, the evidence also shows that in the UK, even when specialised networks have been established, they may not be sustainable. Where this study breaks new ground is that the analysis of the geography of support has previously been absent from the academic literature.

The Forward Cities programme from the US [5] was used as an empirical example of an inclusive entrepreneurial ecosystem that can be built in order to address such equality, diversity and inclusion issues. At issue is sustainability. The Forward Cities example illustrates how fragmentation might be overcome by political will, commitment and sustained investment. In that model, Innovation Councils were set up as formal organisations with a mandate to foster inclusivity. They have performed this by building relationships, social spaces and changing discourses within particular urban spaces. The point here is that this initiative has been sustained and developed by the ongoing commitment of key stakeholders and the incorporation of new ones [6].

This contribution made by this study is in advancing understanding conceptually and empirically of what makes urban and social spaces locally different. An emphasis is on the processes of systemic change through local leadership and advocacy. The conceptual contribution lies in the interpretation of the embeddedness or not of disabled entrepreneurs in urban spaces can be explained by the use of the IEE concept.

In the IEE concept, networks fulfil four system elements of the Stam model: leadership, finance, knowledge and intermediaries, in addition to networking. Two more system elements make up the inclusive entrepreneurial ecosystems approach [4]. These are the role of *advocacy* in driving forward the inclusivity agenda and local and national policy frameworks.

It is these six elements that are used to explain the uneven geography of how minority entrepreneurs belong to and contribute to public spaces. The model differs from the basic Stam model in the key respect that it has, in addition to standard entrepreneurial conditions, fulfilled different functions. These relate to addressing issues of marginality with respect to

inclusion in mainstream business activity, in being connected to organised public urban spaces and in mainstream policy-making.

The networks serve important social functions with respect to the well-being of the disabled and entrepreneurs. Such networks have effects on social systems, social attitudes, on how people see themselves and on what they are capable of doing. They are influenced by how individuals see themselves positioned within society. The case study of the SAMEE project illustrates that where interaction is well-established, minority entrepreneurs add to the cultural richness of places, societal well-being and wealth creation through increasing local equality, diversity and inclusion. The example also illustrates how disabled people need to be both physically and socially embedded in urban spaces.

However, this research shows that very few urban spaces meet this ideal. Differences in local interaction in urban spaces arise from where networks are located. This is fragmented across the UK. In some parts of the UK, there are no networks for a particular group; hence there are no representative organisations that might be leaders and advocates, i.e., agents of change acting on behalf of marginalised entrepreneurs [26].

The IEE model illustrates where the disconnects in the extent to which minority entrepreneurs are included in IEEs in urban spaces. For example, some business organisations do not see minority entrepreneurship as being within their local remit, or support only one kind of minority entrepreneur and not another. Under these circumstances, there is an absence of the adoption of new perspectives [3].

In the UK, the evidence suggests that without such mandates, it is often difficult for potential stakeholders to buy into institutional change or for them to unite around specific objectives and thereby build a dynamic and mutually enforcing environment between a community and interdependent actors that support entrepreneurship. Thus, minority entrepreneurs and their networks remain "marginalised stakeholders" [26]. To be effective, national policy-makers and national business-facing organisations need to recognise local differences, work with what is there and work out how to overcome gaps in provision. This also applies to urban planners who need to take into account the specificities of the physical and social needs of disabled entrepreneurs.

**Funding:** Regional Studies Association, FeRSA Grant May 2019 Round.

**Informed Consent Statement:** Informed consent was obtained from all subjects interviewed in the study.

**Data Availability Statement:** Data can be found in the project report at http://www7.bbk.ac.uk/cimr/2022/09/12/project-report-addressing-regional-inequalities-in-innovation-opportunities-for-ethnically-diverse-and-disabled-entrepreneurs/.

**Acknowledgments:** The author thanks Dina Mansour for her work on data analysis and John Slater for his helpful comments on an earlier version of this paper. The author thanks the Regional Studies Association for funding the research that this paper is based on, Award number FeRSA Grant May 2019 Round.

**Conflicts of Interest:** The author declares no conflict of interest.

## Notes

1    Public Spaces: 10 Principles for Connecting People and the Streets | TheCityFix (accessed on 12 March 2023).

2    Social spaces | Open Green Space (accessed on 5 March 2023).

3    https://thecityfix.com/blog/public-spaces-10-principles-for-connecting-people-and-the-streets-priscila-pacheco/ (accessed on 8 March 2023).

4    https://www.startuploans.co.uk/start-up-loans-partners-with-the-rnib, BEQ_2200029 1..34 (essex.ac.uk) (accessed on 12 March 2023).

5    Disability UK | DISABLED ENTREPRENEUR–DISABILITY UK (accessed on 12 March 2023).

6    https://forwardcities.org/ (accessed on 2 April 2023).

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
