# Peer review of "Public Spaces, Equality, Diversity and Inclusion: Connecting Disabled Entrepreneurs to Urban Spaces"

_land, doi:10.3390/land12040873_

Round 1

Reviewer 1 Report (Previous Reviewer 3)

Dear authors,

Thank you for your interest in submitting the manuscript to the journal Land. I had a chance to review the manuscript that emphasized the social public space.  Based on my reading, I found that the revision was greatly improved, but there are some issues and missing information yet.  

Abstract:
Add the research goal (aim)

Add specific findings from the research.

 Line 18: Remove the abbreviation from the abstract (inclusive entrepreneurial ecosystems (IEE))

Introduction

A capital letter is required for each word, followed by an abbreviation.

Line 23: Entrepreneurs With Disabilities (EWD).

 Line 64: Entrepreneurial Ecosystems (EE)

Line 76: Entrepreneurial Ecosystems (IEE)

Describe the goal of the research clearly in the introduction.  

Analysis

Line 308: (reference)???!!!

3.4 Interview

Based on the first round of comments, I found that my comment was not considered, and the major issue is still found in this section.

For the qualitative data analysis, the author needs to revise this part and presents a detailed process of semi-structured interviews, coding, and thematic analysis, as well as how the findings of the analysis are supported by past findings.

Based on my reading, the main issues are still the lack of a clear analysis process and a lack of evidence to support the interviewees' opinions.

Author Response

The author thanks for the reviewer for their further and helpful comments. The paper has benefitted from this new advice.

Abstract

The research goal has been added;

The goal of the study is to identify what can be done to overcome marginalisation of disabled entrepreneurs which leads to increased local equality of opportunity, thereby adding to the diversity of local economies and thus to a more inclusive society. 

Specific findings have been added:

However, as the evidence from this study of the geography of specialized networks which support disabled entrepreneurs in the UK shows, the entrepreneurial capacity of public spaces (inclusive entrepreneurial ecosystems) for disabled entrepreneurs is better in some places and entirely absent in others.

The abbreviation has been removed.

Introduction

The proper capitalization in each case has been done.

Line 23: Entrepreneurs With Disabilities (EWD).

 Line 64: Entrepreneurial Ecosystems (EE)

Line 76: Entrepreneurial Ecosystems (IEE)

The goal of the research is included.

The goal of the study is to identify what can be done to overcome marginalisation of disabled entrepreneurs which leads to increased local equality of opportunity, thereby adding to the diversity of local economies and thus to a more inclusive society. 

Body of paper

Line 308 reference added British Business Bank, 2020

Interviews

Much more detail has been added on the study and the interview process and analysis.

Two main areas of revision are here:

The new research was carried out in two main stages. The first involved a mapping exercise designed to identify by UK region (including the devolved administrations: Wales, Northern Ireland (NI) and Scotland) all dedicated networks which specifically support ethnic minority, disabled entrepreneurs, or both groups.

The exercise also included identifying all national networks and organisations, as well as government-based initiatives designed to support entrepreneurship and innovation among the two groups.

Various sources and search processes were used to identify networks. These included using a snowball technique of recommendations from participants in the previous study including the Innovate UK project Advisory Board and government agencies. In addition, web searches were conducted to identify networks and data on universities supporting the two target groups was collected. In total 77 organisations were identified 64 of which responded to requests for information and/or interviews.

The second stage involved conducting interviews with networks, individual disabled and ethnic minority entrepreneurs, universities and policy bodies. The sample relating to disability entrepreneurship included 10 networks, 9 policy and/or parliamentary bodies, a national disability organisation, 2 business organisations/trade bodies (e.g. Federation of Small Businesses), 5 academics and 4 other regional bodies (31 in total). Of the ones which did not respond 2 networks specialising in disabled entrepreneurship, two devolved region organisations, four policy making/advisory bodies and two trade bodies.

Computer-based analysis for this research was used using NVivo 12 Plus. The researchers first identified a group of principal categories of data related to the lines of inquiry of the study. This is in line with Strauss and Corbin's (1994) argument that a priori definition of certain baseline concepts in a study supports a grounded theory approach to data gathering and analysis, which is applicable for an interpretive portrayal of the world (Charmaz and Belgrave, 2012).

The a priori codes are codes that are developed before examining the data (interview transcripts). Some of the identified open (initial) codes included:

  • Organisation’s activities and excellence
  • Lack of public policy initiatives
  • Lack of BAME or Disabled support
  • Stakeholder and beneficiary features
  • Effect of location and London centrism

Reviewer 2 Report (New Reviewer)

 The article "Public spaces, equality, diversity and inclusion: connecting disabled entrepreneurs to urban spaces" addresses an important issue of lack of support for disabled entrepreneurs, which is an economic, cultural, and societal issue in the UK and many other countries. The article highlights that disabled entrepreneurs contribute to the cultural richness of places, societal well-being, wealth creation, and increasing local equality, diversity, and inclusion . It draws evidence from a study that focused on the geographies of opportunity and challenges faced by this particular group of entrepreneurs in the UK. The article identifies the agency of particular organizations within inclusive entrepreneurial ecosystems (IEE) as a means of bringing about systemic change, rather than depicting exclusion as embedded within urban spaces. Overall, the article provides valuable insights into the challenges faced by disabled entrepreneurs and the potential benefits of supporting them in urban spaces. The focus on inclusive entrepreneurial ecosystems and the role of organizations in bringing about systemic change is a particularly valuable contribution to the literature on this topic. But I think there are some points in the work which need to be revised.  

- It will be more descriptive if the author(s) present the relationship between " equality" with “diversity ” by triangulations of the main keywords to highlight the importance of the study. There should be a justifiable discussion in the literature part of the study to develop the interrelation. The reason is that I still didn't get the main gap in the study. The relevance of the research problem for the discipline should be highlighted. Why this research is important and how urban planners can get benefit from the findings of this study need to be highlighted.   

-Contribution to academia needs to be highlighted in the abstract, introduction and conclusion part of the study. The contribution of the study needs to be explained in such a way that to increase the originality of the study.

- Introduction doesn’t have any scientific structure to highlight the problem of the study or the gap in the literature. The introduction of the manuscript is not well-organized author may use the strategy of “ big umbrella” to focus on the main problem of the manuscript.  

-In order to increase the internal validity of the study I am highly suggesting to cite the following articles: The Scale of Public Space: Taksim Square in Istanbul. The Role of “Scale” on the Acceleration of Social Interaction in Urban Spaces. Residents’ Social Interactions in Market Square and Its Impact on Community Well-Being.

- The ‘discussion part also needs to develop considering the aim of the article and how the author responds to the hypothesis of the manuscript. The functionality of the methodology and tactics used in the article needs to be discussed.

- The conclusion needs to restructure, some essential information which supposes to be in the conclusion part is missing. For example, what are the findings to support the hypothesis of the study? how the author(s) described the contribution of their study to the existing literature? etc., the Conclusion of the study could be much more descriptive in the findings that the author (s) mentioned on the discussion part.

Author Response

The author thanks the reviewer for their further and very helpful comments. The paper has benefitted from this additional advice.

Comment:

It will be more descriptive if the author(s) present the relationship between " equality" with “diversity ” by triangulations of the main keywords to highlight the importance of the study. There should be a justifiable discussion in the literature part of the study to develop the interrelation. The reason is that I still didn't get the main gap in the study. The relevance of the research problem for the discipline should be highlighted.

Response: The paper now more clearly links equality with diversity by triangulating the main key words to highlight the importance of the study.

For example,

Abstract

The goal of the study is to identify what can be done to overcome marginalisation of disabled entrepreneurs which leads to increased local equality of opportunity, thereby adding to the diversity of local economies and thus to a more inclusive society.  

Introduction

The link to public spaces and equality, diversity and inclusion therefore relates to overcoming marginalisation of disabled entrepreneurs. Where this is addressed by different actors within public spaces, it leads to increased local equality of opportunity, thereby adding to the diversity of local economies and a more inclusive society.

Comment:

Contribution to academia needs to be highlighted in the abstract, introduction and conclusion part of the study. The contribution of the study needs to be explained in such a way that to increase the originality of the study.

Response: The main gaps the study addresses are better spelled out.

Abstract

However, as the evidence from this study of the geography of specialized networks which support disabled entrepreneurs in the UK shows, the entrepreneurial capacity of public spaces (inclusive entrepreneurial ecosystems) for disabled entrepreneurs is better in some places and entirely absent in others. It is this local dimension that has been missing in other studies of disabled entrepreneurs.

Methodology

This project built on evidence from a previous Innovate UK (the UK government’s innovation agency) project (2019) which identified the barriers, challenges, opportunities and support needed for ethnic minority and disabled entrepreneurs (non-gender specific) participating in business innovation (Vorley et al., 2020). The gap in this study was that the geography of support for those groups was largely absent.

Conclusions

Where this study breaks new ground is that the analysis of the geography of support has previously been absent from the academic literature.

Comment

A point is made about the benefit to urban planners

Response: These points are particularly relevant to urban planners who need to take into account the specificities of the physical and social needs of disabled entrepreneurs (page 2).

This also applies to urban planners who need to take into account the specificities of the physical and social needs of disabled entrepreneurs. (conclusions page 17)

Comment

Additional references.

In order to increase the internal validity of the study I am highly suggesting to cite the following articles: The Scale of Public Space: Taksim Square in Istanbul. The Role of “Scale” on the Acceleration of Social Interaction in Urban Spaces. Residents’ Social Interactions in Market Square and Its Impact on Community Well-Being.

Response: Thank you for the additional references. Both have been included in the text.

Comment

‘discussion part also needs to develop considering the aim of the article and how the author responds to the hypothesis of the manuscript. The functionality of the methodology and tactics used in the article needs to be discussed.

Response: These have been addressed in the discussion.  The methodology section is now much more detailed.

Much more detail has been added on the study and the interview process and analysis.

Two main areas of revision are here:

The new research was carried out in two main stages. The first involved a mapping exercise designed to identify by UK region (including the devolved administrations: Wales, Northern Ireland (NI) and Scotland) all dedicated networks which specifically support ethnic minority, disabled entrepreneurs, or both groups.

The exercise also included identifying all national networks and organisations, as well as government-based initiatives designed to support entrepreneurship and innovation among the two groups.

Various sources and search processes were used to identify networks. These included using a snowball technique of recommendations from participants in the previous study including the Innovate UK project Advisory Board and government agencies. In addition, web searches were conducted to identify networks and data on universities supporting the two target groups was collected. In total 77 organisations were identified 64 of which responded to requests for information and/or interviews.

The second stage involved conducting interviews with networks, individual disabled and ethnic minority entrepreneurs, universities and policy bodies. The sample relating to disability entrepreneurship included 10 networks, 9 policy and/or parliamentary bodies, a national disability organisation, 2 business organisations/trade bodies (e.g. Federation of Small Businesses), 5 academics and 4 other regional bodies (31 in total). Of the ones which did not respond 2 networks specialising in disabled entrepreneurship, two devolved region organisations, four policy making/advisory bodies and two trade bodies.

Computer-based analysis for this research was used using NVivo 12 Plus. The researchers first identified a group of principal categories of data related to the lines of inquiry of the study. This is in line with Strauss and Corbin's (1994) argument that a priori definition of certain baseline concepts in a study supports a grounded theory approach to data gathering and analysis, which is applicable for an interpretive portrayal of the world (Charmaz and Belgrave, 2012).

The a priori codes are codes that are developed before examining the data (interview transcripts). Some of the identified open (initial) codes included:

  • Organisation’s activities and excellence
  • Lack of public policy initiatives
  • Lack of BAME or Disabled support
  • Stakeholder and beneficiary features
  • Effect of location and London centrism

Comment

Conclusions

Response: The conclusions have been extensively revised to address the specific points about how the findings support answering the research question and the contribution to the literature.

See for example,

This paper uses data and findings from a recent completed study of network organisations which support disabled entrepreneurs in the UK and from an initiative in the USA. The question raised is How are disabled entrepreneurs embedded in public spaces? At issue is how do urban spaces influence how networks connect disabled entrepreneurs to public urban spaces. This in turn relates to how public spaces can be better sites of equality, diversity and inclusion and diverse social spaces. The goal was to identify what can be done to overcome marginalisation of disabled entrepreneurs which leads to increased local equality of opportunity, thereby adding to the diversity of local economies and a more inclusive society.

The mapping exercise of the geography of support for disabled entrepreneurs in the UK showed considerable variations in the presence of local support in urban places. In some parts of the UK only online support is available. This matters because disabled enrepreneurs may have non-standard physical and social need that require specialised support.

 Of note is evidence from the study of how the agency of networks can lead to overcoming a marginalisation of disabled entrepreneurs. What happens is that in some public places, entrepreneurial capacity is helped by improving inclusive entrepreneurial ecosystems through providing localised specialised support in conjunction with incorporating the interests of local (and national) political and commercial organisations (see for example the SAMEE project example). The network has created ‘a safe space’ for people with disabilities looking to become self-employed.

 In principle agency works on behalf of the marginalised takeholders within public spaces, it leads to increased local equality of opportunity for particular group of entrepreneurs. They thereby add to the diversity of local economies as well as improving the well-being of disabled entrepreneurs (Agboola et al., 2018). However, the evidence also shows that in the UK even when specialized networks have been established, they may not be sustainable. Where this study breaks new ground is that the analysis of the geography of support has previously been absent from the academic literature.

 The Forward Cities programme from the US (Pettit and Pitingolo, 2016) was used as an empirical example of an inclusive entrepreneurial ecosystem that can be built in order to address such equality, diversity and inclusion issues. At issue is sustainability. The Forward Cities example illustrates how fragmentation might be overcome by political will, commitment and sustained investment. In that model, Innovation Councils were set up as formal organisations with a mandate to foster inclusivity.  They have done this by building relationships, social spaces and changing discourses within particular urban spaces. The point here is that this initiative has been sustained and developed by the on-going commitment of key stakeholders and the incorporation of new ones[1].

There are additional changes in the text in the Conclusions.

Round 2

Reviewer 1 Report (Previous Reviewer 3)

Dear author,

After two rounds of reviewing, I found that the revised manuscript addressed the issues.  I read a manuscript that was greatly improved. I have no more questions.

Congratulations!

This manuscript is a resubmission of an earlier submission. The following is a list of the peer review reports and author responses from that submission.

Round 1

Reviewer 1 Report

The article looks at two particular groups of entrepreneurs (disabled and 'ethnic'), the need to support them to perform better, and how this support could provide more consistent inclusivity, local equality and inclusion. The article offers an in-depth study, supported by methods from human geography and economics, describing and discussing the issue, quite precisely, by region in the UK. However, the author is invited to consider (1) a more elaborate definition/discussion of 'public space', as it does not quite fit within this particular article (in which case it will probably be the social public space, depending on the development of the text). (2) Ethnicity, ethnic minorities and disability are different concepts that cover different people, conditions and needs. It would probably be best to consider only one well-defined group to avoid confusion. Within ethnic minorities, the range will be extensive in the UK, with direct implications for their use of public spaces, inclusion and overcoming inequalities.

If the issue is the disability, then health conditions will come to the fore (one might agree that being blind is very different from having a 'learning disability' [BTW - what learning disabilities are there now in the CID medical codes?] A final note on 'lost' links to websites and the line/numbered reference.

I enjoyed reading the paper and welcome its forthcoming publication. 

Reviewer 2 Report

This study examines how specialised networks and organisations for disabled entrepreneurs and ethnic minority entrepreneurs distribute across UK regions, how they function to support the entrepreneurial ecosystem, and how and why their abilities to support this system differ across regions. It is no doubt that the geographic distribution of these networks and the geographic differences in their abilities to support EE are important issues that worth investigation. However, these investigations have little to do with urban spaces in general and public spaces in particular. It seems to me that the author wrote the paper in a way that makes it appear to be linked to the SI if one only looks at its introduction and conclusion, while in fact the rest of the paper has nothing to do with the theme of the IS on public spaces. Therefore, I would suggest the author to reconsider what the paper is really about and submit it to journals with a focus on EE.

The irrelevance of this paper to public spaces can be seen throughout the manuscript. For example, although the author has titled the literature section ‘Place, urban centres, and inclusion’, this section is almost entirely devoted to the IEE concept with no discussion on place and urban centres. Same issues can be found in the empirical section: in 4.4, the author says that what makes the ability to coordinate vary by places is ‘support from local authorities’ – not urban spaces. The paper even does not have any single reference on urban spaces or public spaces. Thus, I do not think the paper is suitable for this SI.

Reviewer 3 Report

Dear authors,

Thank you for your interest in submitting the manuscript to the journal Land.

Reading the manuscript carefully, I found some issues and missing information in it, but the idea is interesting.   

I believe this manuscript does not meet the criteria for Land publication because it does not address novel theoretical arguments, fails to present a convincing link between the theoretical framework and other sections of the manuscript, and lacks data analysis details.

By emphasizing disabled entrepreneurs and ethnically diverse entrepreneurs as an economic, cultural, and societal issue, this study aimed to clarify how urban places affect interactions through specialized networks that act as a conduit for and as a source of support for entrepreneurs.

Despite the fact that UK municipalities are independent and have their own policies regarding space and the urban environment, there is no explanation and no justification for the use of a wide study area for this study. It is imperative to know how the interviewees adjusted their perspectives in this matter. Did they focus on national policies rather than local initiatives to improve inclusion?

I think readers would prefer a case study that focused on a particular city or part of a city rather than the entire UK.

Although they referenced marginalized stakeholder-centric entrepreneurship, the authors did not use that theory to conduct the research in a manner that satisfied the study objectives. The IEE model has been discussed in this work, but there is a major gap between the discussion of theory and the study of urban space. Thus, it is unclear whether this paper significantly advances the existing literature in terms of urban space.

Anyway, you can find my comments as follows:

Abstract

The abstract should be revised to reveal the knowledge gap and/or research problem rather than the research background in the current format.

Introduction

Despite the authors' focus on the significant role of inclusion in urban areas, I cannot see what is the main problem that has not been investigated by any existing studies or research within the field of research in other nationalities. In my reading, the restrictions placed on disabled people to participate in urban spaces are not new, and they have lived with these restrictions for a long time. If you want to establish a bright novelty, you need to show how the current study's findings may fill the knowledge gap in 2023.

Literature Review

I found that section incomplete since the author reviewed urban place rather than an urban space!

Furthermore, the authors neglect the large body of research that has been conducted regarding accessibility, equality for disabled people in urban public areas, or different concepts such as creative cities or talent cities. Due to this, the manuscript could not demonstrate a research gap or how the findings add to what is already known.

Furthermore, the connectivity between the offered model (Entrepreneurial Ecosystem model) and urban space is missing in this section. Due to the lack of a clear theoretical background, it is unclear how research has been conducted to satisfy the objectives. Considering that this is a qualitative study, the author should indicate the type of qualitative research employed.

If the grounded theory has been selected, you need to discuss it.

Method and materials

In this study, qualitative research methods were employed and interviews were conducted to collect data.

It is not clear why the study tried to cover all of the UK in a single research.

Furthermore, I guess the study needs to collect the opinions of architects, urban planners, and urban designers, as well as transportation engineers, but it seems that they are missing.

Results

For the qualitative data analysis, I expected to see a detailed process of semi-structured interviews, rebriefings, coding, and thematic analysis, as well as how the findings of the analysis are supported by past findings.

However, the analysis is not clear and readers cannot recognize “who is who” due to the lack of details. It is not suitable for publication in a high-profile journal such as Land.

Discussion and conclusion

This part is essential for a high-quality paper.

 In addition, please make sure the conclusions section emphasizes the scientific value-added of the paper, and any limitations of current research.

Line 490: Please remove the link from the text and move it to the footnote.

Line 192: kindly remove the underline from the reference (Owen et al., 2015)